# The Relationship between Organisational Factors and Teachers' Psychological Empowerment: Evidence from Lithuania's Low SES Schools

**Loreta Buksnyte-Marmiene** [1,*] , **Agne Brandisauskiene** [2] **and Jurate Cesnaviciene** [3]

1 Department of Psychology, Faculty of Social Sciences, Vytautas Magnus University, Jonavos Str. 66, LT-44191 Kaunas, Lithuania
2 Educational Research Institute, Education Academy, Vytautas Magnus University, K. Donelaicio Str. 52, LT-44248 Kaunas, Lithuania
3 Teacher Training Institute, Education Academy, Vytautas Magnus University, K. Donelaicio Str. 52, LT-44248 Kaunas, Lithuania
* Correspondence: loreta.buksnyte-marmiene@vdu.lt

**Abstract:** Teacher psychological empowerment is one of the main aspects of their effective job performance, job satisfaction and students' higher academic achievement. Unfortunately, there is still little research analysing different organisational factors fostering teacher psychological empowerment. To address this gap, this study asks the following question: how is teacher psychological empowerment associated with organisational factors? The research was performed in 33 schools from 9 municipalities with low SES contexts in Lithuania, and 292 teachers participated in the study. The results of the study show that the school should be viewed as a system in which organisational factors are interrelated and connected with teacher psychological empowerment. It was determined that the purposes of school as organisation predict the general psychological empowerment and teachers' perceived meaning of work. Two organisational factors—purposes and leadership—predict teacher psychological empowerment to make decisions, and teachers' confidence in competence is predicted by three organisational factors: purposes, relationships, and rewards.

**Keywords:** teachers; psychological empowerment; meaning; decision-making; trust in competence; organisational factors; low SES schools





## 1. Introduction

The main tools for successful career and participating in today's global economy is knowledge and quick adjustment to a constantly changing world. This requires teachers to possess the skills and show organisational behaviour which ensure positive learning environment and predict high academic standards for all students. Currently, academics are placing emphasis on an education for sustainable development (further—ESD) and democratic school model in which learner-centred approach and participatory leadership model are the background for teacher behaviour (An and Mindrila 2020). Both ESD and democratic school model are aimed at creating such learning environment that helps to best realize the potential of students. These models emphasize teachers' meaningful leadership in school processes.

UNESCO highlights the need for better teacher leadership skills which are often used as a synonym for teacher empowerment to mitigate learning disparities and support inclusive education at all levels. "The issue of teacher leadership in relation to crisis responses is not just timely, but critical in terms of the contributions teachers have recently made to provide remote learning, support vulnerable populations, re-open schools, and ensure that learning gaps in the curriculum are being mitigated" (UNESCO 2020, p. 1). Therefore, it becomes important for teachers to think independently, effectively solve problems, take responsibilities, cope with tension and dilemmas. Teacher empowerment makes conditions

for such behaviour, teachers' active participation in school life develops their innovative behaviour and supports school development (Celik and Atik 2020). Namely, empowering teachers through autonomy, leadership and opportunities is stressed as one of the most important aspects for teacher career progression (OECD 2019a). However, although teacher empowerment is one of the main strategic goals of ESD and one of the key factors in the achievement of the SDG 4 targets (Nketsia et al. 2020), only 42% of principals report that teachers are significantly involved in making decisions about school policies, curriculum and instruction (OECD 2020). Therefore, uncovering the factors that promote teacher empowerment is significant in both scientific and practical terms.

Research shows that teacher empowerment is one of the main aspects for an effective teacher job performance (Ahmed and Malik 2019; Sharif et al. 2013), job satisfaction (Ahrari et al. 2021; Liu et al. 2021), bigger intrinsic motivation (Oberfield 2016), and students' higher academic achievement (Maniam et al. 2017). It was established that teacher psychological empowerment positively correlates with such organisational behaviour as organisational commitment (Mohammad et al. 2022), professional commitment (Lee and Nie 2014), organisational citizenship behaviour (Macsinga et al. 2015; Saleem et al. 2017) and negatively correlates with a turnover intention (Wijayanti et al. 2020; Ahrari et al. 2021; Tindowen 2019). Moreover, teacher psychological empowerment predicts the individual (cognitive, emotional and intentional) readiness for change (Celik and Atik 2020), which is especially important in nowadays schools. It means that empowered teachers will be more likely to adapt changes and participate in their implementation which is the main strength for managing the challenges created by COVID-19 in schools.

The relationship between the psychological empowerment of teachers and the related consequences has been analysed by scientists, but there is a lack of evidence-based research on the mechanism of the formation of teacher psychological empowerment, the factors that promote/determine greater empowerment of teachers. The scholarly literature has mainly regarded relationship between teacher psychological empowerment and the principal as a leader behaviour (Shah 2014; Elmazi 2018; Gkorezis 2016; Freire and Fernandes 2016) ignoring other organisational factors and processes as important for the formation of empowerment. However, the teacher empowering process cannot develop, as stated by Kang et al. (2021), inside a vacuum. Thus, it is important to analyse the school as a system and to reveal the role of organisational factors of teacher psychological empowerment. It should be noted that low-SES (socioeconomic status) schools must also become the object of these studies. Research claims that the psychological empowerment of teachers significantly contributes to the academic achievement and learning success of students from low socioeconomic status (SES) (Maniam et al. 2017).

Thus, this study asks the following question: How is teacher psychological empowerment associated with organisational factors such as the school's purposes, structure, leadership, relationships, rewards, helpful mechanisms, and attitude toward the change in low-SES school context?

## 2. Theoretical Background

### 2.1. The Phenomenon of Teacher Psychological Empowerment

Teacher empowerment is a multidimensional phenomenon and is treated in various ways. Different types of empowerments are distinguished, such as societal, structural, organisational, psychological etc., as well as different structural parts of the empowerment concept. In this study, we will analyse the psychological teacher empowerment.

Psychological empowerment is derived from Bandura's (1977) self-efficacy theory. This concept was further developed by Conger and Kanungo (1988), who described empowering as a motivational process in which individuals develop their self-efficacy. This idea was expanded by Thomas and Velthouse (1990), who claim that empowerment should be understood as a multidimensional structure consisting of a combination of four elements: meaning, self-determination, competence, and impact. These authors claim that psychological empowerment provides energy to a person's behaviour and causes a person's intrinsic

motivation towards one's work role, which obviously creates conditions for better job performance. Spreitzer's (1995) argues similarly, supporting the four-dimensional (impact, competence, meaning and self-determination) structure of psychological empowerment and claiming that psychological empowerment is employees' perception of how much they control their work environment. Separately, each dimension of psychological empowerment (spot of the structure) also has its own meaning. Meaning dimension means that employees who feel empowered believe that their work is meaningful. Competence means that empowered employees tend to feel that they are capable of performing their jobs effectively. Impact reflects employees' perception on their ability to influence the decision making and the processes that take place in the organisation. Self-determination refers to employees' awareness of their ability to initiate and regulate their own work activities. It enables employees to feel powerful (Spreitzer 1995; Thomas and Velthouse 1990). The listed aspects of the empowerment structure are covered by the phenomenon of self-determination, which shows how much one or other behaviour depends on the employee themself and not external conditions (Tvarijonavičius et al. 2016). Therefore, it can be stated that psychological empowerment is closely related to self-determination theory (Ryan and Deci 2000) and can be associated with meeting the needs of autonomy, relatedness, competence. Researchers (Yildiz et al. 2017) note that psychological empowerment promotes teacher autonomy, greater involvement in decision-making, a sense of control in relation to their work and feeling of trust toward both themselves and their organisation.

In general, psychological empowerment includes the teachers' experience of mastery and motivational energy, and can be associated with positive attitudes, behaviours, and performance. It is the teachers' perception of how much they want (meaning dimension), are able (decision-making) and know how to (confidence in ones' competence) successfully perform what is expected of them at work (Tvarijonavičius et al. 2016). In other words, psychological empowerment refers to the way employees experience their work and their personal perception about their role in relation to the organisation (Ambad and Bahron 2012). According to researchers, psychological empowerment can be used as a tool to motivate teachers and to increase their level of performance in teaching and research (Sotirofski 2014). An employee who feels psychologically empowered feels freedom to make choices while fulfilling a duty (Celik and Atik 2020), and also tries to improve the performance by working "smarter" or by seeking out new and better ways of doing things (Fernandez and Moldogaziev 2013). Thus, psychological empowerment promotes proactive rather than passive attitude to one's work roles (Kim and Lee 2020).

In the educational context, teacher empowerment is defined as a process that encourages the teacher to be involved in decision-making, expands their decision-making capabilities and trust in them as a decision-maker, encourages taking responsibility and gives a sense of control over the process (Ahrari et al. 2021; Yildiz et al. 2017). A psychologically empowered teacher is characterized by greater autonomy, responsibility, belief in their competences and application of them in work practice, ability to teach their students effectively (Muhammad and Hussain 2020; Shah 2014). They believe and care about what they do, are more satisfied, engaged and innovative (Yildiz et al. 2017). It is the empowerment of teachers that makes better use of the school's intellectual resources to foster student achievement, which is especially important when schools are under-resourced.

It should be noted that psychological empowerment should not be perceived as a static phenomenon or a characteristic of a person, but as a process in which an employee (teacher) can feel more or less psychologically empowered. This process can be influenced both by the school principals' behaviour and by other processes taking place in the workplace (a school) as well as in the organisation. Psychological empowerment may vary with organisational structure, individual and team characteristics, work design, leadership, and organisational support (Kim and Lee 2020). The teachers' personal perceptions towards their work environment are also important in this process (Yildiz et al. 2017). As Peist

et al. (2020) states, power is fluid and teachers may simultaneously be empowered and disempowered in different contexts.

Summarizing, it can be noted that although individual authors' definitions of psychological empowerment differ slightly, most authors agree and identify the same essential components of this phenomenon: the meaning of work for the employee, the perceived opportunity to independently solve work issues and personal responsibility for decisions and trust in one's professional competence. Therefore, in this study of Lithuanian teachers, we used the Lithuanian Psychological Empowerment Questionnaire, the revised version, which includes the dimensions of meaning, decision-making and confidence in competence. This questionnaire is validated and adapted for the Lithuanian sample (Tvarijonavičius et al. 2016).

### 2.2. The Significance of the Teachers' Psychological Empowerment for Their Behaviour and Job Outcomes

Research shows that teacher psychological empowerment is a significant factor that has consequences in two directions: the organisational behaviour of teachers and the learning of students. When assessing the significance of teacher psychological empowerment for organisational behaviour, it was found that greater psychological empowerment is related to better job performance, with passion for working and higher work results both directly (Sanli 2019; Yu and Kim 2021; Ahmed and Malik 2019) and indirectly by influencing employee self-efficacy, motivation and job satisfaction (Fernandez and Moldogaziev 2013). Moreover, psychological empowerment at work is significantly associated with well-being of the employees and organisations (Macsinga et al. 2015) and negatively associated with teacher burnout (Tsang et al. 2022; Kaya and Altınkurt 2018). The results of the conducted research reveal the importance of psychological empowerment of teachers in their increased intrinsic motivation (Yildiz et al. 2017), greater teacher job satisfaction, professional commitment and the decision to remain in the profession (Lee and Nie 2014; Shen et al. 2012; Wijayanti et al. 2020; Ahrari et al. 2021; Tindowen 2019). As stated by Berry et al. (2010), by psychologically empowering teachers, schools can not only improve the quality of teaching, but also retain the most effective teachers, which may be especially relevant for schools that face certain challenges (e.g., low-SES schools).

Teacher psychological empowerment is also significantly related to such organisational behaviour as organisational commitment (Mohammad et al. 2022; Lee and Nie 2014), organisational citizenship behaviour (Macsinga et al. 2015; Saleem et al. 2017; Aksel et al. 2013), and engagement in work (Tindowen 2019); in addition, it increases trust in school principal (Freire and Fernandes 2016). Research shows that the psychological empowerment of teachers is associated with a more active desire of teachers to implement the school's goals and take care of the more successful functioning of the school as an organisation (Yu and Kim 2021; Elmazi 2018), more active cooperation with colleagues and participation in school community activities (Tindowen 2019).

In terms of student learning, the psychological empowerment of teachers is related to their interaction with students, student behaviour and their achievements: empowered teachers seek new ideas and aim to implement them in their teaching practice, tend to experiment and take risks by challenging their comfort zone, more actively monitor student progress and provide feedback and change their teaching strategy to meet the needs of their students (Tannehill and MacPhail 2017; Shah 2014; Celik and Atik 2020). Such teachers experience feelings of professional confidence and pride, which promote effective teaching practices, the desire to become an even better teacher, and concern for student progress (Nazari et al. 2021; Tannehill and MacPhail 2017). These teachers are committed to their students, devoting more time and energy to preparing and teaching students, focusing on their well-being and achievement. It should be noted that teacher empowerment is a crucial aspect for schools with large numbers of low SES students. Researchers point out that students from low-SES may be characterised by lower academic achievement (OECD 2019b), lower motivation to learn (Erentaitė et al. 2022), less confidence in their

own development, i.e., they have a fixed mindset (Brandisauskiene et al. 2022; Wang et al. 2021) and more behavioural problems and mental health difficulties (Patalay et al. 2020). Therefore, teacher empowerment is a crucial aspect for schools with large numbers of low-SES students.

Finally, the psychological empowerment of teachers is related to their approach to change, their behaviour in implementing it: an empowered teacher is in favour of innovation, is ready for change and tends to demonstrate innovative behaviour (Gkorezis 2016; Yildiz et al. 2017; Gil et al. 2018). Thus, psychological empowerment is the key to the success of educational reforms and a significant factor in the formation of effective tools for the effective implementation of changes in schools. This research finding is particularly important in the context of COVID-19 and can be leveraged as a strength to help schools cope with the challenges that have arisen in the context of the pandemic.

In summary, the research reveals a significant role of teacher psychological empowerment in school functioning, teacher quality of performance and student learning success. As Shah (2014) stated, teachers possess knowledge, experience and understanding of the classroom realities and their input on various professional issues help to improve functioning of a school as an organisation. In other words, teacher empowerment correlates with successful school improvement and reform because it creates "a critical mass of empowered experts within the building" (Berry et al. 2010, p. 4).

### 2.3. The Role of the Organisational Factors for Fostering Teacher Psychological Empowerment

Previous studies by researchers from other countries emphasize the importance of the leadership and organisational processes for teacher empowerment (e.g., Kang et al. 2021; Kiral 2020; Yildiz et al. 2017). Principal behaviour is reported as one of the critical factors for teacher psychological empowerment (Kang et al. 2021). For example, it has been observed that distributed leadership is an effective leadership style in promoting teacher psychological empowerment, while hierarchical leadership style is related to teacher disempowerment (Shah 2014). Previous studies predominantly analyse the significance of a certain leadership style in teacher psychological empowerment, while in this study we ask teachers' opinions about leadership in their school, the degree to which leadership is useful, helping the organisation to develop and achieve the set goals and the significance of this phenomenon for teacher empowerment.

Teacher communication with the school principal is mentioned as an important organisational factor affecting teacher psychological empowerment. As stated by Yao et al. (2020), empowerment itself (power sharing) requires communication between leaders and members; moreover, only when employees perceive communication as fully empowering them do they experience a higher sense of meaning, competence, self-determination, and impact. Freire and Fernandes (2016) research results show that access to information, resources, support and opportunity contribute to the psychological empowerment of teachers. According to these scientists, the psychological empowerment of teachers has a positive effect on their trust of the school principal. Thus, interpersonal working relationships affect teacher psychological empowerment. The research results show that the lack of collaboration and support negatively influence teacher psychological empowerment, while institution and educator greater engagement in intra and inter collaboration foster the psychological empowerment of educators (Zeb et al. 2019). The study of Khany and Tazik (2016) confirms this regularity—trust in principal and colleague is indirectly related to job satisfaction through teacher psychological empowerment. Peist et al. (2020) also names the relational component as one of the essentials for the psychological empowerment of teachers and claims that breakdowns in communication are central to teacher disempowerment. In addition, based on the results of this research, it is emphasized that, besides the communication, important factors of teacher disempowerment are lack of support and inconsistent application of school rules by the leadership. Thus, effective principals enhance teacher psychological empowerment and at the same time the credibility and confidence in the organisation (Elmazi 2018). Although leadership and communication are analysed

as important antecedents of teacher empowerment, there is still a lack of evidence-based research on the way in which such organisational factors as relationship between colleagues, rewards and other helpful mechanisms are associated with teacher empowerment. Moreover, when analysing the interaction of some of the above-mentioned organisational factors with employee empowerment, ambiguous results are obtained. For instance, although the results of the Freire and Fernandes (2016) study confirm the importance of organisational support in teacher empowerment, no analogous relationships were found in the sample with employees from manufacturing firms (Kumar et al. 2022). It can be considered that the significance of organisational factors in employee empowerment also depends on the nature of the activities performed, therefore the results of previously conducted research cannot be summarised/generalised to all research samples, and the study of psychological empowerment of teachers working in SES-context schools is meaningful because it reveals regularities specific to this sample.

Another important factor for teacher psychological empowerment is organisational culture. This organisational factor and its individual components greatly affect the psychological empowerment of the educators (Zeb et al. 2019). The researchers claim that poor organisational structure (they distinguish this as a component of organisational culture) and unclear description of educator role and responsibilities is an obstacle to teacher psychological empowerment. A study conducted in academicians sample revealed that organisational culture has a great impact on the psychological empowerment: although hierarchy culture is dominant in universities, the strongest predictor of psychological empowerment is clan culture (Sotirofski 2014). The study author emphasizes that open communication and flexibility as characteristics of organisational culture as well as environment which is more friendly to the employees rather than controlling them are very important factors of psychological empowerment of employees. It is evident that a supportive culture is very important for the psychological empowerment of teachers in their job, which manifests itself not only through the school leader, but also the respect and care of colleagues (Kang et al. 2021). Collaboration and support are also mentioned as important organisational factors in other studies. Shah (2014) notes that lack of collaboration and interpersonal trust negatively affects teacher psychological empowerment. Although it is possible to find research that analyses organisational culture in general, there is a lack of research on the aspects of organisational culture such as clearly communicating organisational purposes to employees, encouraging discussion about purposes and commitment to them, the effect of organisational structure and flexible division of labour on teachers' feeling of empowerment. In addition, the question arises as to how the organisation's attitude toward change could be associated with teacher empowerment. This becomes an especially important issue during times of change such as the COVID-19 pandemic.

In summary, the conducted research suggests that organisational factors can make an impact on teacher psychological empowerment. According to Bogler and Nir (2012), organisational leaders can positively affect their subordinates and enhance their empowerment through organisational processes. Based on the analysis of the literature and previous research and aiming to fill the gap in research, in this study, we hypothesize that such organisational factors as purposes, structure, leadership, relationships, rewards, helpful mechanisms and attitude change predict the teacher psychological empowerment.

## 3. Materials and Methods

### 3.1. Research Participants

The research data were collected in May 2021. A stage sampling procedure was used. At the first stage, thirty-three secondary education schools were selected from 9 municipalities with low-SES contexts in Lithuania. These schools are small and located in small towns or rural areas. Then, all the teachers at the school sample were invited to participate in this study. Teachers who expressed interest in participating received an email with a consent form and a link to the online questionnaire. In total, 292 teachers accepted the invitation and voluntarily participated in the study (34 men and 258 women). The rate

of participation was 53.7%. Out of the teachers who participated in the study, 22.6% were beginning teachers working at school for less than 5 years; 18.8% of teachers had 6–15 years of teaching experience; almost a quarter of teachers (24.7%) had 16–25 years of teaching experience, and one third of teachers (33.9%) reported that they had more than 25 years of teaching experience.

### *3.2. Data Collection Instruments*

The anonymous self-reported questionnaire consisted of three sections. Section one included the demographic information to describe participants: gender and years of teaching experience. The second part was intended to determine the teachers' opinions about organisational factors (i.e., purpose, structure, relationships, rewards, leadership, helpful mechanisms, and attitude towards change). The Organisational Diagnosis Questionnaire by Preziosi (1980) was used for this. The questionnaire is composed of 7 subscales and 35 items (5 statements for each organisational factor):

purposes—the research subjects assess whether the organisation goals are clearly defined, whether they are understood by the employees, whether the employees are involved in setting purposes and agree with them;

structure—the research subjects assess whether the structure of the organisation and department is suitable for effective goal achievement, whether the division/distribution of work in the organisation is logical and flexible in helping to achieve goals;

leadership—the research subjects evaluate whether the leadership is useful, helping the organisation to develop and achieve the set goals, as well as the degree to which the employees receive the support from the leader;

relationships—the research subjects evaluate the strength and quality of relationships with colleagues, as well as the presence/absence of unresolved conflicts in the organisation;

rewards—the research subjects evaluate the objectivity of the reward system based on its impartiality and successful functioning in the organisation;

helpful mechanisms—the research subjects evaluate whether helpful mechanisms such as the dissemination of information in the organisation, planning, control, the help of departments for each other aid in employees' job performance and development of the organisation;

attitude toward change—the subjects assess the degree to which the organisation is open and favourable to change, promotes innovation, and is able to change.

Each item was being rated on a seven-point Likert scale ranging from disagree strongly (1) to agree strongly (7). The higher composite score of the response indicated a better assessment of a specific organisational factor. Reliability of the Organisational Diagnosis Questionnaire was measured by Cronbach's alpha which was reported to be 0.961. Table 1 presents item examples and Cronbach's α estimates for each subscale.

**Table 1.** Reliability of the questionnaires.

| Subscales | Cronbach α | Number of Items | Sample of Items |
|---|---|---|---|
| Organisational Diagnosis Questionnaire (α = 0.961) | | | |
| Purpose | 0.801 | 5 | I am personally in agreement with the stated goals of my work unit |
| Structure | 0.841 | 5 | The division of labour in this organisation actually helps it to reach its goals |
| Relationships | 0.717 | 5 | I have established the relationships that I need to do my job properly |
| Rewards | 0.800 | 5 | My job offers me the opportunity to grow as a person |
| Leadership | 0.714 | 5 | This organisation leadership efforts result in the organisation's fulfilment of its purposes |
| Helpful mechanisms | 0.844 | 5 | Other work units are helpful to my work unit whenever assistance is requested |
| Attitude toward change | 0.842 | 5 | Occasionally I like to change things about my job |
| Lithuanian Employee Psychological Empowerment Questionnaire (α = 0.879) | | | |
| Meaning | 0.728 | 3 | My work is meaningful to me |
| Decision-making | 0.767 | 3 | I can make my own decisions at work |
| Trust in competence | 0.788 | 3 | I think that other people at work trust my competences |

In section three, teacher psychological empowerment was assessed using the Lithuanian Employee Psychological Empowerment Questionnaire (LPEQ–9) developed by Tvarijonavičius et al. (2016). Each of the empowerment dimensions (i.e., meaning, decision-making, trust in competence) was measured by three items. The response scale was a six-point Likert scale ranging from 1 (strongly disagree) to 6 (strongly agree). The three sub-scales average indexes were formed for both, meaning that the indexes also ranged between 1 and 6. The higher the composite score of the response, the higher the teacher perception of being more psychologically empowered. Reliability of the Lithuanian Employee Psychological Empowerment Questionnaire, as assessed by Cronbach's alpha, was 0.879. Table 1 demonstrates Cronbach's α estimates for each psychological empowerment dimension.

### 3.3. Data Analysis

The statistical analyses were performed using IBM SPSS Statistics 26.0. We used descriptive statistics, Pearson bivariate (zero-order) correlation, ANOVA with post hoc Tukey's multiple comparison tests, and multiple regression analysis. Reliability of the measuring instruments examined by calculating Cronbach's alpha. The Cronbach's alpha of 0.70 or higher for a set of items is considered acceptable (Cohen et al. 2018). Statistical significance was set at $p < 0.05$.

In our study, the independent and dependent variables are measured with one self-reported questionnaire. Therefore, responses of research participants could have common method bias (CMB). In order to test if the collected data are prone to CMB, Harman's single-factor test was conducted. The cut-off point for the current test is 50% variance (Podsakoff et al. 2003). The test result shows that the variance for the first factor is 39.69%, indicating that the influence of the CMB on the statistical results is not significant.

## 4. Results

### 4.1. School Organisational Factors

Table 2 presents descriptive statistics of organisational factors. Mean, median, mode, skewness, and kurtosis values indicate that the distribution of the data is close to a normal distribution. Judging from the mode values, all seven organisational factors were rated high by the teachers who participated in the study (mode is equal to 6). However, looking at the minimum value and average, it can be seen that teachers rated one of the organisational factors—rewards—worse, and purpose—the best.

**Table 2.** Descriptive statistics of organisational factors.

| Subscales | Min | Max | Mean | SD | Median | Mode | Skewness | Kurtosis |
|---|---|---|---|---|---|---|---|---|
| Purpose | 4.0 | 7 | 6.05 | 0.62 | 6 | 6 | −0.677 | 0.538 |
| Structure | 3.4 | 7 | 5.89 | 0.75 | 6 | 6 | −0.847 | 0.693 |
| Relationships | 3.6 | 7 | 6.01 | 0.66 | 6 | 6 | −0.795 | 0.720 |
| Rewards | 2.6 | 7 | 5.52 | 0.96 | 6 | 6 | −0.677 | 0.071 |
| Leadership | 3.4 | 7 | 5.84 | 0.75 | 6 | 6 | −0.833 | 0.631 |
| Helpful mechanisms | 3.2 | 7 | 5.83 | 0.82 | 5.6 | 6 | −0.855 | 0.487 |
| Attitude towards change | 3.6 | 7 | 6.03 | 0.70 | 6 | 6 | −0.976 | 1.145 |

Table 3 presents positive statistically significant strong and very strong correlation between all organisational factors (at $p < 0.01$ level).

In order to determine the way in which teachers with different working experiences evaluate organisational factors, the ANOVA test was applied. The obtained results show that there are no statistically significant differences, i.e., assessment of school organisational factors does not depend on the teachers' years of teaching experience (Table 4).

**Table 3.** Pearson's bivariate correlation between seven organisational factors.

| Subscales | 1 | 2 | 3 | 4 | 5 | 6 | 7 |
|---|---|---|---|---|---|---|---|
| (1) Purpose | - | | | | | | |
| (2) Structure | 0.803 ** | - | | | | | |
| (3) Relationships | 0.752 ** | 0.797 ** | - | | | | |
| (4) Rewards | 0.682 ** | 0.796 ** | 0.684 ** | - | | | |
| (5) Leadership | 0.682 ** | 0.709 ** | 0.660 ** | 0.676 ** | - | | |
| (6) Helpful mechanisms | 0.808 ** | 0.876 ** | 0.770 ** | 0.792 ** | 0.771 ** | - | |
| (7) Attitude towards change | 0.784 ** | 0.808 ** | 0.741 ** | 0.683 ** | 0.706 ** | 0.842 ** | - |

** Correlation is significant at the 0.01 level (2-tailed).

**Table 4.** Comparison of evaluation of school organisational factors by years of teaching experience.

| Organisational Factors | Years of Teaching Experience | Mean | SD | ANOVA Test Results | |
|---|---|---|---|---|---|
| | | | | F | *p* |
| Purpose | less than 5 years | 6.07 | 0.56 | 1.822 | 0.143 |
| | 6–15 years | 6.14 | 0.62 | | |
| | 16–25 years | 5.91 | 0.61 | | |
| | more than 25 years | 6.08 | 0.66 | | |
| Structure | less than 5 years | 5.98 | 0.76 | 1.284 | 0.280 |
| | 6–15 years | 5.91 | 0.80 | | |
| | 16–25 years | 5.75 | 0.75 | | |
| | more than 25 years | 5.94 | 0.69 | | |
| Relationships | less than 5 years | 6.10 | 0.66 | 2.548 | 0.056 |
| | 6–15 years | 6.08 | 0.65 | | |
| | 16–25 years | 5.83 | 0.66 | | |
| | more than 25 years | 6.03 | 0.66 | | |
| Rewards | less than 5 years | 5.52 | 0.91 | 1.165 | 0.323 |
| | 6–15 years | 5.71 | 0.96 | | |
| | 16–25 years | 5.45 | 0.96 | | |
| | more than 25 years | 5.43 | 0.98 | | |
| Leadership | less than 5 years | 5.88 | 0.75 | 0.280 | 0.840 |
| | 6–15 years | 5.90 | 0.80 | | |
| | 16–25 years | 5.81 | 0.82 | | |
| | more than 25 years | 5.80 | 0.66 | | |
| Helpful mechanisms | less than 5 years | 5.92 | 0.84 | 1.236 | 0.297 |
| | 6–15 years | 5.91 | 0.85 | | |
| | 16–25 years | 5.68 | 0.82 | | |
| | more than 25 years | 5.83 | 0.79 | | |
| Attitude towards change | less than 5 years | 6.06 | 0.65 | 0.721 | 0.540 |
| | 6–15 years | 6.01 | 0.76 | | |
| | 16–25 years | 5.93 | 0.77 | | |
| | more than 25 years | 6.09 | 0.66 | | |

*4.2. Teacher Psychological Empowerment*

Descriptive statistics of the three dimensions of teacher psychological empowerment (meaning, decision-making, trust in competence) and general psychological empowerment are presented in Table 5. The results show that the distribution of the data is close to the normal distribution. The evaluation of all three dimensions of psychological empowerment and the expressiveness of overall psychological empowerment is quite high (mode equal to 5). These results reveal that some teachers perceive their work as meaningful, feel that they can influence the decisions made, trust their competence and believe that other members of the school community also trust their competence. In other words, teachers feel relatively high psychological empowerment. However, it is necessary to pay attention to the fact that the decision-making dimension was rated worse by some teachers (minimum score 1.67).

**Table 5.** Descriptive statistics of the teachers' psychological empowerment.

|  | Min | Max | Mean | SD | Median | Mode | Skewness | Kurtosis |
|---|---|---|---|---|---|---|---|---|
| Meaning | 3 | 6 | 5.09 | 0.61 | 5 | 5 | −0.533 | 0.376 |
| Decision-making | 1.67 | 6 | 4.62 | 0.76 | 4.67 | 5 | −0.495 | 0.578 |
| Trust in competence | 3.33 | 6 | 4.96 | 0.58 | 5 | 5 | −0.246 | 0.055 |
| Psychological empowerment | 3.33 | 6 | 4.89 | 0.57 | 4.89 | 5 | −0.173 | −0.054 |

The ANOVA test results show that the evaluation of two dimensions of psychological empowerment (meaning and trust in competence) and general psychological empowerment do not depend on the teachers' years of teaching experience (Table 6). However, it turned out that the psychological empowerment of the teachers who participated in the study differed in decision-making. After applying the post hoc Tukey HSD test, it was determined that the average of psychological empowerment to make decisions of teachers who have been working for more than 25 years (M = 4.76) is statistically significantly higher than the average (M = 4.39) of teachers with teaching experience of less than 5 years (F = 3.159, $p < 0.05$). In other words, teachers who have been working for more than 25 years feel more empowered to make decisions independently, to influence the decisions made and other members of the school community than teachers with less than 5 years of experience in the school.

**Table 6.** Comparison of the teachers' psychological empowerment by years of teaching experience.

| Organisational Factors | Years of Teaching Experience | Mean | SD | ANOVA Test Results | |
|---|---|---|---|---|---|
|  |  |  |  | F | p |
| Meaning | less than 5 years | 5.05 | 0.62 | 0.283 | 0.837 |
|  | 6–15 years | 5.07 | 0.62 |  |  |
|  | 16–25 years | 5.10 | 0.56 |  |  |
|  | more than 25 years | 5.13 | 0.63 |  |  |
| Decision-making | less than 5 years | 4.39 | 0.91 | 3.159 | 0.025 |
|  | 6–15 years | 4.63 | 0.72 |  |  |
|  | 16–25 years | 4.63 | 0.67 |  |  |
|  | more than 25 years | 4.76 | 0.71 |  |  |
| Trust in competence | less than 5 years | 4.92 | 0.63 | 0.615 | 0.606 |
|  | 6–15 years | 5.05 | 0.50 |  |  |
|  | 16–25 years | 4.94 | 0.55 |  |  |
|  | more than 25 years | 4.95 | 0.62 |  |  |
| Psychological empowerment | less than 5 years | 4.79 | 0.63 | 1.078 | 0.359 |
|  | 6–15 years | 4.92 | 0.52 |  |  |
|  | 16–25 years | 4.89 | 0.53 |  |  |
|  | more than 25 years | 4.95 | 0.58 |  |  |

### 4.3. Relationships between School Organisational Factors and Teacher Psychological Empowerment

The correlation matrix (Table 7) has been shown positive and significant relationship between all variables (at $p < 0.01$ level). Dimensions of teacher psychological empowerment and overall psychological empowerment are statistically significantly related to all organisational factors. It should be noted that teacher psychological empowerment and all three of its dimensions are most closely related to the purposes of the school as an organisation: a statistically significant relationship of moderate strength was established with meaning (r = 0.509), decision-making (r = 0.476), trust in competence (r = 0.502) and psychological empowerment (r = 0.566).

**Table 7.** Pearson's bivariate correlation between organisational factors and teacher psychological empowerment.

|  | Purpose | Structure | Relationships | Rewards | Leadership | Helpful Mechanisms | Attitude Towards Change |
|---|---|---|---|---|---|---|---|
| Meaning | 0.509 ** | 0.398 ** | 0.387 ** | 0.324 ** | 0.358 ** | 0.430 ** | 0.446 ** |
| Decision-making | 0.476 ** | 0.439 ** | 0.444 ** | 0.384 ** | 0.438 ** | 0.459 ** | 0.462 ** |
| Trust in competence | 0.502 ** | 0.393 ** | 0.437 ** | 0.293 ** | 0.348 ** | 0.437 ** | 0.428 ** |
| Psychological empowerment | 0.566 ** | 0.473 ** | 0.486 ** | 0.388 ** | 0.443 ** | 0.508 ** | 0.512 ** |

** Correlation is significant at the 0.01 level (2-tailed).

### 4.4. Organisational Factors Predicting Teacher Psychological Empowerment

A multiple regression was run to find out how organisational factors predict teacher psychological empowerment. Due to multicollinearity of independent variables (VIF > 4), the following independent variables had to be removed from multiple regression: structure, helpful mechanisms, and attitude towards change. The remaining independent variables were included in the regression analysis: purpose, relationships, rewards, and leadership.

First of all, the way in which organisational factors predict the general psychological empowerment of teachers was investigated. The resulting regression model explains 33.5% of the variance (F = 36.102, $p < 0.0001$). However, in this model (Table 8), there is only one statistically significant predictor—purpose. After repeating the linear regression with only this predictor, the coefficient of determination decreased slightly ($R^2 = 0.321$; F = 137.049, $p < 0.0001$). It shows that 32.1% of the total psychological empowerment of teachers is predicted by the purposes of the school as an organisation, and the linear regression equation is written as follows:

$$\text{psychological empowerment} = 1.753 + 0.519 \text{ purpose.} \tag{1}$$

**Table 8.** Organisational factors predicting teacher psychological empowerment.

|  | Unstandardized Coefficients | | Standardized Coefficients β | t | p |
|---|---|---|---|---|---|
|  | B | Std. Error |  |  |  |
| Constant | 1.512 | 0.288 |  | 5.252 | 0.0001 |
| Purpose | 0.413 | 0.074 | 0.450 | 5.558 | 0.0001 |
| Relationships | 0.117 | 0.068 | 0.137 | 1.719 | 0.087 |
| Rewards | −0.048 | 0.044 | −0.081 | −1.087 | 0.278 |
| Leadership | 0.076 | 0.056 | 0.100 | 1.372 | 0.171 |

When analysing the research results, it was important to find out the way in which organisational factors predict each dimension of psychological empowerment separately. As shown by correlational analysis, perception of meaning, i.e., the teachers' belief in what they do at work, perception of the meaningfulness of their work and positive attitude towards work are positively related to organisational factors. Multiple regression helped to determine which of them can predict it. The resulting regression model explains 26.1% of the variance (F = 25.379, $p < 0.0001$), but from the results we can see that there is again only one statistically significant predictor—the purpose of the school as an organisation (Table 9).

After improving the regression model, i.e., leaving only a statistically significant predictor, the obtained coefficient of determination slightly decreases—$R^2 = 0.259$ (F = 101.425, $p < 0.0001$). Thus, 25.9% of the meaning dimension is predicted by the purposes of the school as an organisation. The final linear regression equation is written as the following:

$$\text{Meaning} = 2.051 + 0.503 \text{ purpose.} \tag{2}$$

Next, the way organisational factors predict the psychological empowerment of teachers to make decisions is explained. The resulting regression model explains 25.8% of the variance (F = 24.977, *p* < 0.0001). In this model (Table 10), statistically significant predictors are purpose and leadership.

**Table 9.** Organisational factors predicting the teachers' perception of meaning.

| | Unstandardized Coefficients | | Standardized Coefficients β | t | *p* |
|---|---|---|---|---|---|
| | B | Std. Error | | | |
| Constant | 1.961 | 0.327 | | 5.989 | 0.0001 |
| Purpose | 0.504 | 0.084 | 0.510 | 5.971 | 0.0001 |
| Relationships | 0.021 | 0.078 | 0.023 | 0.273 | 0.785 |
| Rewards | −0.044 | 0.050 | −0.068 | −0.867 | 0.387 |
| Leadership | 0.034 | 0.063 | 0.042 | 0.541 | 0.589 |

**Table 10.** Organisational factors predicting the empowerment of teachers to make decisions.

| | Unstandardized Coefficients | | Standardized Coefficients β | t | *p* |
|---|---|---|---|---|---|
| | B | Std. Error | | | |
| Constant | 0.742 | 0.408 | | 1.821 | 0.070 |
| Purpose | 0.314 | 0.105 | 0.255 | 2.984 | 0.003 |
| Relationships | 0.159 | 0.097 | 0.138 | 1.645 | 0.101 |
| Rewards | −0.001 | 0.063 | −0.002 | −0.021 | 0.983 |
| Leadership | 0.177 | 0.079 | 0.173 | 2.250 | 0.025 |

After leaving only statistically significant predictors in the regression model and repeating the multiple regression, the coefficient of determination obtained almost does not change ($R^2$ = 0.251; F = 48.357, *p* < 0.0001). We find that purpose and leadership predict 25.1% of the way teachers perceive their ability to make decisions independently, to influence decisions and other people. The final linear regression equation is written as the following:

$$\text{decision-making} = 0.892 + 0.409 \text{ purpose} + 0.215 \text{ leadership.} \tag{3}$$

Finally, the way organisational factors predict teacher confidence in competence was tested. The resulting regression model explains 27.2% of the variance (F = 26.808, *p* < 0.0001). In this model, there are three statistically significant predictors: purpose, relationships, and rewards (Table 11).

**Table 11.** Organisational factors predicting the empowerment of teachers to make decisions.

| | Unstandardized Coefficients | | Standardized Coefficients β | t | *p* |
|---|---|---|---|---|---|
| | B | Std. Error | | | |
| Constant | 1.831 | 0.307 | | 5.957 | 0.0001 |
| Purpose | 0.423 | 0.079 | 0.452 | 5.337 | 0.0001 |
| Relationships | 0.170 | 0.073 | 0.194 | 2.330 | 0.020 |
| Rewards | −0.099 | 0.047 | −0.163 | −2.094 | 0.037 |
| Leadership | 0.017 | 0.059 | 0.022 | 0.290 | 0.772 |

Leaving only statistically significant predictors in the regression model, the obtained coefficient of determination did not change ($R^2$ = 0.272, F = 35.830, *p* < 0.0001). Thus, it was found that such organisational factors as purpose, relationships, and rewards predict 27.2% of the teachers' perception that they are competent to perform their work properly and achieve results. The final linear regression equation is written as follows:

$$\text{trust in competence} = 1.845 + 0.429 \text{ purpose} + 0.174 \text{ relationships} - 0.095 \text{ rewards.} \tag{4}$$

Summarizing the results of multiple regression, it can be said that not all organisational factors predict teacher psychological empowerment and its dimensions. It turned out that only purpose predicts overall psychological empowerment and the teachers' perceived meaning of work. Two organisational factors—purpose and leadership—predict teacher psychological empowerment to make decisions, and the teachers' confidence in competence is predicted by three organisational factors: purpose, relationships, and rewards.

## 5. Discussion

In order to achieve the successful psychological empowerment of teachers, it is important to analyse the factors affecting this phenomenon. In this study, we focused on the role of the organisational factors for fostering teacher psychological empowerment in low-SES context schools in Lithuania. To our knowledge, such a study of the relationship between organisational factors and teacher psychological empowerment is the first in Lithuania. The results of the conducted research allow us to identify several important aspects.

First, Lithuanian teachers from low-SES-context schools who participated in the study claim that they feel a relatively high general psychological empowerment in their job. This means that they perceive their work as meaningful, believe in what they do at work, have a positive attitude towards work, feel that they can make decisions independently in various work situations, influence decisions, perceive themselves as competent to do their work properly, can overcome difficulties and achieve the desired results. As other researchers claim, these characteristics increase the teachers' effectiveness (Ahmed and Malik 2019; Sharif et al. 2013), teacher job satisfaction and well-being (Bogler and Nir 2012), reduce the likelihood of teacher burnout (Tsang et al. 2022) and are particularly significant when working with students in low-SES-context schools (Maniam et al. 2017). However, it should be noted that, although teachers rated the expressiveness of all three dimensions of psychological empowerment similarly, teachers rated the dimension of decision-making as less expressive, and the dispersion of responses in this dimension was the highest compared to the other two dimensions of psychological empowerment. The research data show that there were also teachers who rated the decision-making dimension as poorly expressed (the lowest rating was 1.67 out of 6 possible). Thus, based on the results, we can say that the teachers who participated in the study feel the least empowered in terms of independent decision-making and the ability to choose. The teachers express an ambiguous experience in relation to decision-making—some feel that they can independently decide and choose in work situations, others believe that they do not have the opportunity to choose, because the behaviour in work decision-making situations does not depend on themselves, but rather on external conditions. It is possible that this reflects the attitude of teachers sometimes heard in public discussions in Lithuania that a lot of decisions are handed down to the schools of our country "from above" and that teachers feel unable to positively impact the overall educational system (Poteliūnienė et al. 2019). It is interesting that it is only in this decision-making dimension that differences in the expression of teacher psychological empowerment were found when comparing groups of teachers with different years of teaching work experience (no differences were found in other dimensions of psychological empowerment in this regard): teachers with more than 25 years of teaching experience in the school feel significantly more empowered in decision-making dimension compared to young teachers with up to 5 years of work experience in the school. We would encourage future research to investigate this phenomenon in more detail and answer the question of whether these differences reflect the principals' varying behaviour towards more and less experienced teachers, giving the former more authority and freedom to make decisions independently compared to other teachers, or whether it reflects the teachers' internal attitudes, when gaining more and more teaching work experience and, along with it, expert experience, enables teachers to make decisions bolder and more actively. The need for more detailed research on this issue is also inspired by the fact that no unequivocal answer to the question of the way psychological empowerment is related to teacher work experience

and other sociodemographic characteristics has been found in the studies so far (Kiral 2020; Veisi et al. 2015; Kang et al. 2021).

Second, the research results confirm our assumption that the organisation (school) should be viewed as a system in which all organisational factors are related/interwoven and interact with each other. The results obtained in this study show strong intercorrelations between all analysed organisational factors. It can be assumed that if one organisational factor does not function in the school, it affects other organisational factors as well, and vice versa. Thus, we would invite both researchers and practitioners seeking the schools' improvement to take this into account.

Third, in order to answer the question of how teacher psychological empowerment is associated with organisational factors, we found that all 7 studied organisational factors are related to teacher psychological empowerment in general and individual dimensions of psychological empowerment. This confirms the thoughts of other researchers (e.g., Zeb et al. 2019) about the importance of organisational factors for teacher psychological empowerment and suggests that it would be more appropriate to analyse teacher psychological empowerment not at the micro, but at the macro level, taking into account the school organisational factors as an important context, promoting, facilitating psychological empowerment process or hindering it (Lee and Nie 2014). In addition, it also reflects the vision that the processes taking place in the school should be viewed as a whole, strengthening the complex integrated perspective of different organisational factors' impact on teacher psychological empowerment (Kang et al. 2021).

However, analysing deeper the significance of organisational factors for teacher psychological empowerment, it was found that teacher psychological empowerment in general and the meaning dimension are most predicted by the purposes of the school as an organisation. This organisational factor is significant for predictor and decision-making as well as trust in competence dimensions. This means that the more positively teachers assess the organisation's goals, the better these goals are known, clear and acceptable to them, the more actively teachers are involved in formulating the goals of the school as an organisation, the more empowered they feel. This is especially important for the teachers' understanding of the meaning of their job. Analysing the significance of organisational factors for the decision-making and trust in competence dimensions, it was found that, besides purposes, another significant predictor for the decision-making dimension is leadership, while trust in competence without purposes also significantly predicts relationships and rewards. The obtained research results correspond to that regularity observed in the studies of other researchers: the principal's behaviour and communication with teachers is resiliently related to the teachers' psychological empowerment (Kang et al. 2021). For example, Yao et al. (2020) found that communication between principal and teachers when a principal understands and accepts teachers' emotions, ideas, values, conveys information related to teachers' work, asks teachers' opinion, encourages and supports teachers can significantly and positively predict psychological empowerment. Thus, leadership is an organisational factor that empowers employees if the leader trusts them and shares the responsibility for achieving organisational goals (Shah 2014). Attention should also be paid to the importance of relationships between colleagues for the psychological empowerment of teachers: positive, cooperative and supportive relationships between colleagues are considered a significant predictor for the teachers' trust in competence.

In summary, the conducted research allows us to state the important links between the schools' organisational factors and teacher psychological empowerment. The significant contribution of this study to the research on the phenomenon of teacher psychological empowerment is that it was found that the prognostic value of organisational factors for individual dimensions of psychological empowerment is different. Researchers (e.g., Lee and Nie 2014) claim that different dimensions of psychological empowerment are differently related to the teachers' work-related outcomes, so our study expands this research field by substantiating the significance of different organisational factors as antecedents for different dimensions of teacher psychological empowerment. The task for future research

is to answer the following question: To what extent do the regularities identified in the study reflect the characteristics of teachers working in low-SES-context schools, and to what extent is this characteristic of the general population of teachers?

Finally, it is necessary to discuss the limitations of this study. First, our research design was cross-sectional. This makes it possible to evaluate the correlations between organisational factors and teacher psychological empowerment but does not allow for a more detailed analysis of psychological empowerment as a process, the features of its formation, and the significance of organisational factors for that. Therefore, we would suggest choosing a longitudinal research design for future studies. Secondly, the very positive evaluations of teachers in relation to the analysed phenomena raise the question of the social desirability bias problem and the teachers' desire to give higher evaluations than what they would evaluate the phenomena in reality. On the other hand, it is possible to consider the assumption that those teachers who felt more empowered than those who did not agree to participate in the study agreed to participate in the study. However, there are more studies in which teachers rate them as having a highly expressed psychological empowerment (e.g., Tindowen 2019; Sanli 2019), so the assumption that our study reflects the real assessment of teachers cannot be ruled out. However, in order to avoid encountering this problem in future studies, it would be worthwhile to use different research data collection sources (e.g., not only from teachers, but also principals, social partners, etc.). Third, all measures in our study are based on self-reports. This means that the results may be affected by common method bias. We controlled this by performing an exploratory factor analysis, i.e., Harman's single-factor test. In future research, we would suggest choosing not only self-reports, but include some objective or observable variables.

However, while acknowledging the limitations of the study, we do not want to lose sight of the result of the study, namely that in the studied Lithuanian schools where teachers work with students from low SES there is an emerging tendency of teachers feeling empowered. This is a joyful result, because one of the main documents which describes school conception in Lithuania—the Good School Concept—emphasizes the empowerment of teachers, the participation of all members of the school community in decision-making as an important aspect of increasing the quality of education (The Good School Concept 2015). It is likely that empowered teachers become active creators of the school as an organisation, maintaining a democratic school and aiming for education for sustainable development, which is especially meaningful when working in low-SES schools. Based on the results of our research, we would recommend schools in low-SES context to formulate their purposes as an organisation very clearly, to agree on them with the entire school community, and to promote collaboration between teachers. Recognizing that teacher empowerment can be identified as one of the key variables to improve school performance (Jiang et al. 2019), our study can serve as a basis for further research on this phenomenon in low-SES schools.

## 6. Conclusions

The results of the conducted research allow us to say that the school should be viewed as a system in which organisational factors are intertwined and interact with each other. This is shown by the established strong intercorrelations between all analysed organisational factors such as purposes, structure, leadership, relationships, rewards, helpful mechanisms and attitude toward change. These organisational factors are also related to teacher psychological empowerment in general as well as individual dimensions of psychological empowerment. Analysing the significance of organisational factors for the psychological empowerment of teachers, it becomes clear that the psychological empowerment of teachers in general and the meaning dimension are most predicted by the purposes of the school as an organisation. This organisational factor is significant predictor for decision-making and trust in competence dimensions. The teachers' decision-making is also predicted by the school principal's behaviour (leadership), and the teachers' trust in competence is influenced by the relationships between teachers and the principal, as well as the system of rewards. The result of the study, namely that the prognostic significance of

organisational factors for individual dimensions of psychological empowerment differs, is important and encourages further research.

**Author Contributions:** Conceptualization, L.B.-M.; methodology, L.B.-M. and J.C.; investigation, L.B.-M. and A.B.; formal analysis, L.B.-M. and J.C.; writing—original draft preparation, L.B.-M.; writing—review and editing, L.B.-M. and A.B. All authors have read and agreed to the published version of the manuscript.

**Funding:** This article has received funding from European Social Fund (project "Comprehensive University Development in the Context of Universities Network Restructurisation", No 09.3.1-ESFA-V-738-02-0001).

**Institutional Review Board Statement:** The study was conducted in accordance with the Declaration of Helsinki and approved by the Ethics Committee of Education Academy Vytautas Magnus University, Lithuania (Protocol number: SA-EK-21-03; 20 April 2021).

**Informed Consent Statement:** Informed consent was obtained from all subjects involved in the study.

**Data Availability Statement:** Not applicable.

**Conflicts of Interest:** The authors declare no conflict of interest. The funders had no role in the design of the study; in the collection, analyses, or interpretation of data; in the writing of the manuscript; or in the decision to publish the results.

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
