# Peer review of "The Relationship between Organisational Factors and Teachers’ Psychological Empowerment: Evidence from Lithuania’s Low SES Schools"

_socsci, doi:10.3390/socsci11110523_

Round 1
Reviewer 1 Report
Very relevant and interesting research.
Author Response
We would like to thank a lot for the time that the Editor and the reviewers have spent on reading our manuscript and provided meaningful suggestions to improve it further.
Reviewer’s 1 comment |
Authors’ answer |
Very relevant and interesting research. |
We would like to thank a lot for the time You have spent on reading our manuscript. |
Reviewer 2 Report
The topic of the research (the relationship between teachers' psychological empowerment and organisational factors) is interesting and the research can potentially be a relevant contribution to the research field. However, there are major and minor serious problems in the paper on the basis of which I decided to reject the paper. Below I will describe these problems:
· There is no alignment between the literature on organisational factors and the instrument used. The summary on page 6 is not a summary, it introduces various new concepts such as purposes, structure, rewards, helpful mechanisms, and attitudes towards change. How do they relate to the concepts described in the introduction, such as communication and organisational culture?
· The authors should have used multi-level analysis, as organisational factors surely vary according to schools.
· The study has been conducted in low-ses schools and the authors indicate this is a relevant context factor, however, they do not explain why and how, and they cite just one paper that supports the relevance.
· The organisational factors that are measured were not introduced (see remark above) and therefore the results are hard to understand. For example: ‘leadership’ is measured, but what is the meaning of the scale? Does it refer to distributed or hierarchical leadership as described in the introduction?
· What is the response rate of schools and of teachers within schools? To what extent is the sample a representation of the teacher workforce in Lithuania?
· Why is years of experience relevant? It may be, but given the central role in the results section, the relevance should have been explained.
· About the multicollinearity of the organisational factors: how did you arrive at the decision to include the variables you did (top of page 8).
· The quality of the use of English is quite poor (many of the articles are missing for example), the quality of the text in general needs much improvement. There is a lot of repetition and thinking steps that are missing in arguments (for example in the first paragraph: there is a jump from ESD to teachers as facilitators which is not explained).
· The word ‘influence’ in the title is not justified, given the cross-sectional nature of the data.
Author Response
We would like to thank a lot for the time that the Editor and the reviewers have spent on reading our manuscript and provided meaningful suggestions to improve it further. Your comments allowed us to take a new look at our research and deepened our experience in writing manuscript. We appreciate your efforts to make our manuscript better. Thank you very much. The changes which we made in the manuscript regarding reviewers’ comments are marked in the text via track change.

Reviewer 3 Report
A well designed and executed research study. However, there needs to be made thorough reference to the limitations inherent in it .i.e. what was the teacher population of the schools included in the study? what was the rate of participation? the sample was limited to teachers participating on a voluntary basis; this may have significant implications for the results of the study etc.
Author Response
We would like to thank a lot for the time that the Editor and the reviewers have spent on reading our manuscript and provided meaningful suggestions to improve it further. Your comments allowed us to take a new look at our research and deepened our experience in writing manuscript. We appreciate your efforts to make our manuscript better. Thank you very much. The changes which we made in the manuscript regarding reviewers’ comments are marked in the text via track change.
Reviewer’s 3 comment |
Authors’ answer |
A well designed and executed research study. However, there needs to be made thorough reference to the limitations inherent in it .i.e. what was the teacher population of the schools included in the study? what was the rate of participation? the sample was limited to teachers participating on a voluntary basis; this may have significant implications for the results of the study etc. |
We appreciate your comment. We invited 57 schools to take part in the survey and 33 schools agreed to participate (the rate of participation – 57.9%). These schools are small and located in small towns or rural areas with low SES contexts. All teachers in these schools were invited to take part in the study. 292 teachers volunteered to take part in the study (the rate of participation – 53.7%). In response to your observation, we have supplemented a methodology section of the manuscript. |
Reviewer 4 Report
I appreciated the opportunity to read your research entitled “The Influence of Organisational Factors on Teachers’ Psychological Empowerment: Evidence from Lithuania’s Low SES Schools”. I applaud the attempt to investigate organisational factors (i.e., purposes, structure, leadership, relationships, rewards, helpful mechanisms and attitude toward change) on the teachers' psychological empowerment. However, I have several concerns and recommendations about the paper;
· Paper is over-generalized and doesn’t reflect novelty. According to the authors’ knowledge, such a study of the relationship between organisational factors and teachers' psychological empowerment is the first in Lithuania. It only fills the contextual gaps where organizational factors are taken to the surface. There are different studies available targeting specific factors (i.e., bureaucratic structure of schools, perceptions of executive/organizational support, ethical leadership, etc.) on specific outcomes which are directly and indirectly associated with one’s psychological empowerment. Hence, it’s strongly recommended to highlight the importance of the study clearly, not only by demonstrating the contextual gaps but how it differs from others. As such, why authors have chosen organizational factors in general? How does this study contribute practically to low SES schools? Which organizational factor these low SES schools should consider to empower teachers in Lithuania?
· Article is too wordy and roams around the empowerment by repeatedly citing old references multiple times in a single sentence (i.e., P4/L190 Macsinga et al. 2015; Tindowen 2019). Please concise the text, avoid constant repetition of references and cite new work (recommended below).
· P3/L134-35: Recent study (2022) confirms empowered employees’ proactive behavior (taking charge) under organizational support. Cited work (theoretical overview) is last modified in 2016 and doesn’t support the argument.
· There is a big difference between the male-female ratio. How do authors see the categorical variable as a gap? Both have different personality traits and experiences; their empowerment activation could be different for diverse organizations at different hierarchal levels.
· How is teachers’ psychological empowerment associated with organizational factors? This is the main question the study is based on. The authors ended up asking self-reported questions and using Harman’s single-factor test for CBM (outdated). If the study intends to truly empower teachers of Lithuania in low SES schools, it shouldn’t rely on single-reported sources, and that too in the generalized form. There can be different leadership styles, reward systems, knowledge sharing or hoarding behaviour, organizational cynicism, relationship conflict etc. Why would anyone self-report about him/herself that I hoard knowledge? Because of poor LMX, one might say that leadership is poor, and many other concerns.
Kock, N., 2015. Common method bias in PLS-SEM: A full collinearity assessment approach. International Journal of e-Collaboration (ijec), 11(4), pp.1-10.
· Discussion is lengthy with unnecessary text and repeated sentences from the introduction and literature review. It needs to be more concise and specific with past studies supporting the present study’s outcomes. Also, this study’s theoretical and practical implications.
Recommended literature;
- Investigation of the relationship between teachers’ perceptions of executive support, employment, school effectiveness, and job satisfaction (2020)
- Evaluation of Employee Empowerment on Taking Charge Behaviour: An Application of Perceived Organizational Support as a Moderator (2022)
- Exploring the Direct and Indirect Influence of Perceived Organizational Support on Affective Organizational Commitment (2022)
- The relationship between the bureaucratic structure of schools, organisational silence, and organisational cynicism (2016)
- How ethical leadership influences professional learning communities via teacher obligation and participation in decision making: A moderated-mediation analysis (2020)
- The impact of ethical leadership on organisational citizenship behaviours: Moderating role of organisational cynicism
I hope my comments will help you to improve the content. Good luck!
Author Response

(The authors gave the same response as above.)

Round 2
Reviewer 4 Report
Thanks for putting effort and incorporating recommendations.